# A Quantitative Evaluation of the Effectiveness of the Metal Artifact Reduction Algorithm in Cone Beam Computed Tomographic Images with Stainless Steel Orthodontic Brackets and Arch Wires: An Ex Vivo Study

**DOI:** 10.3390/diagnostics14020159

**Published:** 2024-01-10

**Authors:** Mojgan Shavakhi, Parisa Soltani, Golnaz Aghababaee, Romeo Patini, Niccolò Giuseppe Armogida, Gianrico Spagnuolo, Alessandra Valletta

**Affiliations:** 1Department of Orthodontics, School of Dentistry, Isfahan University of Medical Sciences, Isfahan 8174673461, Iran; shavakhi.m@gmail.com; 2Department of Oral and Maxillofacial Radiology, Dental Implants Research Center, Dental Research Institute, School of Dentistry, Isfahan University of Medical Sciences, Isfahan 8174673461, Iran; p.soltani@dnt.mui.ac.ir; 3Department of Neurosciences, Reproductive and Odontostomatological Sciences, University of Naples “Federico II”, 80138 Naples, Italy; ng.armogida@gmail.com (N.G.A.); alessandra.valletta@unina.it (A.V.); 4Students Research Committee, School of Dentistry, Isfahan University of Medical Sciences, Isfahan 8174673461, Iran; aghababaeegolnaz@gmail.com; 5Department of Head, Neck and Sense Organs, Università Cattolica del Sacro Cuore, Fondazione Policlinico Universitario A. Gemelli, 00168 Rome, Italy; romeo.patini@unicatt.it; 6Therapeutic Dentistry Department, Institute for Dentistry, Sechenov University, 119991 Moscow, Russia

**Keywords:** cone beam computed tomography, orthodontic brackets, metal artifact, metal artifact reduction

## Abstract

The presence of high-density and high-atomic number materials results in the generation of artifacts in cone beam computed tomographic (CBCT) images. To minimize artifacts in CBCT images, the metal artifact reduction (MAR) tool was developed. This study aims to quantitatively evaluate the effectiveness of the MAR algorithm in CBCT images of teeth with stainless steel orthodontic brackets with or without arch wires in buccal and lingual positions obtained using the Galileos Sirona CBCT scanner. In this in vitro study, 20 stainless steel brackets were attached to the maxillary dentition from the right second premolar to the left second premolar teeth of a human skull. In the first group, 10 brackets were bonded to the buccal surface, and in the second group, 10 brackets were bonded to the palatal surface of these teeth. CBCT scans were obtained for each group with or without orthodontic stainless steel wires using a Galileos Sirona CBCT scanner with exposure parameters of 85 kVp and 21 mAs. CBCT images were obtained two times with and two times without MAR activation. The DICOM format of the CBCT images was imported to ImageJ software (version 1.54), and the contrast-to-noise ratio (CNR) was calculated and compared for each bracket in 15 and 20 mm distances and 20, 40, and 90 degrees on each side. Statistical analysis was performed using the *t* test (α = 0.05). CNR values of different distances and different teeth were not significantly different between the two MAR modes (*p* > 0.05). MAR activation had a significant impact in increasing CNR and reducing artifacts only when brackets were in palatal (*p* = 0.03). In the other bracket and wire positions, the effect of the MAR algorithm on CNR was not significant (*p* > 0.05). In conclusion, MAR activation significantly increased CNR, but only when the brackets were in a palatal position. In the other bracket and wire positions, the effect of the MAR algorithm is not significant.

## 1. Introduction

Treatment with fixed orthodontic appliances is the preferred method for the treatment of most malocclusions, and the most common components used in it are brackets, tubes, band materials, ligating materials, and arch wires [1].

Orthodontic brackets can be made of metal, such as stainless steel and titanium, or non-metallic materials, such as plastic, ceramic, and plastic composite [2]. Moreover, orthodontic brackets can be bonded to the labial or lingual surface of the teeth [3]. Lingual orthodontic treatment has entered mainstream practice due to its esthetic advantages over conventional labial orthodontics [4]. In addition, lingual appliances have been shown to decrease the risk of enamel decalcification compared to labial brackets and offer high-precision outcomes [5].

In the past, panoramic and lateral cephalometric radiographs were considered standard imaging modalities for the evaluation of orthodontic patients. However, these radiographs are limited by their two-dimensional nature and thus cannot effectively visualize the relationships of different structures. Additionally, as a result of geometric magnification and/or distortion, measurements in these conventional radiographs are not accurate in all regions of the image [6]. Cone beam computed tomography (CBCT) imaging provides the possibility of obtaining high-resolution three-dimensional information of hard tissue with a relatively low radiation dose. CBCT imaging overcomes the major disadvantages and limitations of conventional radiographs. Therefore, the use of CBCT is very helpful in many dental fields, including orthodontics [7]. In the field of orthodontics, CBCT is used for the evaluation of impacted and ectopic teeth, the assessment of airways, the evaluation of mini-implant sites, the analysis of craniofacial deformities, the preoperative planning of orthognathic surgery, and the evaluation of root resorption, among other applications [8].

One of the disadvantages of CBCT images is the presence of artifacts [9]. Any error or distortion in radiographic images that is not related to the subject under investigation is called an artifact. CBCT image artifacts are divided into four categories based on their etiology: (1) inherent artifacts, which are artifacts that are caused by the inherent characteristics of CBCT imaging; e.g., cone beam geometry, voxel size, and area detector; (2) procedure-related artifacts, which are artifacts arising during image acquisition; (3) introduced artifacts, which are artifacts resulting from the contents of the field of view; and (4) patient motion artifacts [10].

Objects with high density and atomic number can cause metal artifacts in CBCT images, which are classified among introduced artifacts [11]. When X-rays pass through an object, photons with low energy are absorbed more than photons with high energy [12]. This phenomenon is called beam hardening, which causes two types of artifacts: (1) distortion in metal structures as a result of differential absorption of photons, also known as a cupping (blooming) artifact, and (2) dark stripes and lines that are between two metal objects, which are called extinction or missing value artifacts [12]. These artifacts reduce the quality of CBCT images and distort the image of structures around metallic objects [13,14].

In order to reduce these artifacts and increase the quality of CBCT images, metal artifact reduction (MAR) algorithms have been used. These algorithms are applied to the images in the reconstruction of tomographic images and reduce or, if possible, completely eliminate the artifacts [11]. However, their effectiveness for different diagnostic tasks has yet to be investigated. Cebe et al. evaluated the effect of different restoration materials on the diagnosis of proximal caries with and without MAR activation mode. Their findings indicated that these algorithms increase the accuracy of caries detection [15]. Kim et al. prepared CBCT images with and without MAR from four phantoms with prostheses made of amalgam, gold, zirconia, and porcelain fused to metal, with different scanning parameters. Based on their results, MAR reduces the amount of streaked artifacts, but its effect depends on the settings of the device and the type of prosthesis [16]. The negative impact of MAR algorithms on diagnosis has also been reported. For instance, Oliveira et al. investigated the effect of MAR on the diagnosis of vertical root fracture and concluded that MAR reduces the accuracy of vertical root fracture diagnosis [17]. As shown by these conflicting results, the influence of metal artifact reduction algorithms is task-specific and dependent on a variety of different factors. In the field of orthodontics, Isman et al. showed that the presence of orthodontic wires and brackets prevented the diagnosis of proximal caries in CBCT images. The application of MAR or limiting the field of view did not influence the diagnosis of interproximal caries lesions [18]. Additionally, McLaughlin et al. reported that the application of the Planmeca MAR algorithm did not affect the diagnosis of contact between an anchorage device and teeth [19].

To the authors’ knowledge, no quantitative study has been performed on the performance of CBCT MAR algorithms in the reduction in artifacts caused by brackets and arch wires located in buccal and lingual positions. Particularly, the comparison of the amount of artifacts and the performance of MAR algorithms when brackets and wires are located in the lingual position has not been explored. Therefore, the primary objective of this study was to quantitatively evaluate the effectiveness of the MAR algorithm in CBCT images of teeth with SS orthodontic brackets in buccal and lingual positions obtained using the Galileos Sirona CBCT scanner. The secondary aim was to assess the effect of the insertion of the orthodontic arch wires on the amount of artifacts in each condition.

## 2. Materials and Methods

This in vitro study was approved by the Research Ethics Committee of Isfahan University of Medical Sciences (IR.MUI.RESEARCH.REC.1401.040).

### 2.1. Sample Preparation

In this study, an unidentified human skull with intact maxillary teeth without caries was selected from the Department of Oral and Maxillofacial Radiology, Isfahan University of Medical Sciences, Iran. First, CBCT images were prepared from teeth without orthodontic wires and brackets. Then, for the first group of images, 10 stainless steel metal brackets (American orthodontics MBT 022, Sheboygan, WI, USA) were bonded to the buccal surface of maxillary teeth, from the right second premolar to the left second premolar. For the second group of images, buccal brackets were de-bonded, and 10 new stainless steel brackets were bonded to the palatal surface of the same teeth. To attach the brackets, the enamel surface was first etched with 37% phosphoric acid (American orthodontics, Sheboygan, WI, USA) for 30 s, washed with water for 30 s, and then the tooth was dried with air flow. Bonding (Brace paste; American orthodontics, Sheboygan, WI, USA) was applied for 30 s and thinned with gentle air flow and light cured for 30 s. Then, adhesive (Brace paste; American orthodontics, Sheboygan, WI, USA) was applied and light-cured for 30 s after placing the bracket. In both groups, CBCT images were prepared once without wire and once with stainless steel wire (American orthodontics (0.016 × 0.022 in.), Sheboygan, WI, USA). All the stainless steel wires were attached to the brackets by O-rings (American orthodontics, Sheboygan, WI, USA). To simulate the soft tissue and also to provide a homogenous material for the quantification of artifacts, the skull was placed in a container of water.

### 2.2. CBCT Scan

In total, the different positions of brackets and wires bonded to the skull were as follows: (1) skull without bracket and wire (No-BW); (2) skull with brackets attached to the buccal surface (B-B); (3) skull with brackets and wire attached to the buccal surface (BW-B); (4) skull with brackets attached to the palatal surface (B-P); and (5) skull with brackets and wire attached to the palatal surface (BW-P). CBCT images were obtained for each of these conditions twice without activation of the MAR and then twice with MAR activation. We performed two CBCT scans for each MAR mode to eliminate the variability of the scans. All images were obtained using a Galileos CBCT scanner (Sirona, Bensheim, Germany) with exposure parameters of 85 kVp and 21 mAs with a voxel size of 280 μm. Reconstruction of the CBCT images was performed using Sidexis software (version 4, Sirona, Bensheim, Germany).

### 2.3. Quantitative Analysis of CBCT Images

The DICOM format of the CBCT images was imported into ImageJ software (version 1.54, NIH, Bethesda, MD, USA). For each bracket, A fixed axial image was selected as a reference with the help of a marker placed in the middle level of the brackets, and all measurements were performed on this axial image. The measurements were performed by a trained senior dental student under the supervision of an oral radiologist.

At first, a line was drawn from the center of the bracket along the buccal surface of the tooth. A line perpendicular to this line was drawn, and 10 regions of interest (ROIs) of the same size (2 mm × 2 mm) were established for brackets bonded to the right central incisor to the right second premolar. ROIs were located at intervals of 15 and 20 mm from the center of the bracket and with angles of 20 and 40 degrees on each side of the perpendicular line and also on the perpendicular line (Figure 1). The mean and standard deviation (SD) of gray values were calculated for each ROI. To calculate the contrast-to-noise ratio (CNR), an ROI with similar dimensions was selected in the water surrounding the skull, and the mean and SD of gray values were considered controls. The CNR is calculated as follows:$$CNR=\frac{\left|{Mean}_{bracket}-{Mean}_{control}\right|}{\sqrt{{SD}_{bracket}^{2}+{SD}_{control}^{2}}}$$

Therefore, in each CBCT image, 10 CNRs were calculated for each bracket: 5 CNRs for 15 mm and 5 CNRS for 20 mm distances. Then, the CNRs for each distance were averaged. Thus, for each bracket, two CNR values were achieved: one in 15 mm distance and one in 20 mm distance to the center of the bracket.

Data were entered into Statistical Package for the Social Sciences (SPSS, version 25, IBM Statistics, New York, NY, USA). Statistical analysis was performed using a *t* test (α = 0.05).

## 3. Results

Overall, there were no significant differences between the average CNR values in the two MAR modes (*p* = 0.220, Table 1). Additionally, the mean CNR values of the ROIs in 15 and 20 mm distances from the center of the brackets were not significantly different between the on and off MAR conditions (*p* = 0.657 and *p* = 0.092, respectively, Table 2).

Moreover, for each tooth, the mean CNR values between the on and off MAR modes were not significantly different (*p* > 0.05, Table 3).

For the B-P skull, the mean CNR values in the on MAR mode were significantly higher than in the off mode (*p* = 0.030). However, for the other bracket and wire positions, the difference between the CNR values in the two modes was not statistically significant (*p* > 0.05, Table 4).

## 4. Discussion

According to the results of this study, CNR values of different distances and for different teeth were not significantly different between the two MAR modes. But, activating MAR reduces metallic artifacts only when the brackets are in a palatal position.

There are various clinical conditions that make imaging necessary during orthodontic treatments. For example, maxillary canine teeth are one of the most common teeth that are impacted [20,21]. The orthodontic treatment plan for impacted canine teeth can include tooth extraction or exposure of the tooth and applying force to it to be placed in the dental arch [20]. In both of these cases, accurate diagnosis of the features of the impacted tooth, including its position, the condition of the dental follicle, and its effects on the adjacent structures, is of particular importance [22]. Additionally, in cases where orthodontic brackets and wires are connected to teeth, various imaging techniques for other diagnostic purposes may be prescribed.

With the increasing use of CBCT as the most widely used 3D modality in dentistry, the application of this technique in the diagnosis and orthodontic treatment plan of maxillary impacted canines has been considered. CBCT allows for the examination of structures in three dimensions and with high resolution with a relatively low radiation dose [7]. However, one of the challenges of CBCT imaging is the presence of metallic artifacts around objects with a high atomic number [23]. For example, in cases where metallic orthodontic brackets are attached to the surface of adjacent teeth, due to their high atomic number, artifacts are created around them. In such cases, accurate detection of details such as the presence of root resorption in teeth, the presence of a lesion, and fracture of the adjacent bone is challenging [24,25].

Gray values are the basis of digital radiographs and can be used to quantitatively evaluate the features of tissues and structures [26,27]. In an effort to reduce metallic artifacts, different manufacturers have developed different algorithms that can replace the gray values affected by the artifacts with the actual values and thus decrease the effect of artifacts on structures. In order to determine the usefulness of these algorithms, it is necessary to study their efficiency in different diagnostic cases. One of the possible applications of these algorithms is to enhance the CBCT image of individuals with orthodontic wires and brackets. Based on the search conducted by the authors, no study has been previously conducted to quantitatively measure the effect of metal artifact reduction algorithms on CBCT images of teeth with stainless steel brackets and wires. However, regarding other diagnostic tasks, numerous studies have been conducted on the effectiveness of MAR algorithms.

In 2017, Cebe et al., in a study, put 98 molar and premolar teeth that had proximal caries in contact with teeth that were restored with amalgam, modified glass ionomer, resin composite, ceramic composite, and zirconia and evaluated the effect of different restoration materials on the diagnosis of proximal caries with and without MAR activation and concluded that these algorithms increase the accuracy of caries detection [15]. The CBCT machine and the MAR algorithm used in the study of Cebe et al. were from Planmeca company, whereas the CBCT scanner and MAR algorithm from Sirona company were used in the present study.

In 2020, in a study by Fontenele et al. CBCT scans were obtained from a human mandible once with a zirconia implant and once without it in three MAR modes (without MAR, MAR before exposure, and MAR after exposure). Artifacts were assessed using SD of gray values and CNR in six ROIs with different distances (10 to 35 mm) and angulations (70 to 135) from the implants. The authors concluded that both modes of using MAR reduce the amount of artifacts in CBCT images, especially when the impact of artifacts in the images is more significant [23]. This finding is not completely in line with our findings. In our study, the only instance where MAR was effective in reducing the artifacts was when the brackets were placed in the palatal position. The reason can be attributed to different metallic materials and MAR algorithms used in the two studies.

In another study in 2020, Kim et al. prepared CBCT images with and without MAR from four phantoms with prostheses made of amalgam, gold, zirconia, and porcelain fused to metal, as well as with settings of 100 and 70 kVp and voxel size of 0.2 and 0.3 mm. They concluded that MAR reduces the amount of streaked artifacts, but its effect depends on the settings of the device and the type of prosthesis [16]. The current study did not focus on the role of exposure settings and its main purpose is the effect of the position of the brackets and the presence or absence of wires on the performance of the artifact reduction algorithm.

In 2021, Oliveira et al. investigated the effect of MAR on the diagnosis of vertical root fracture and concluded that MAR reduces the accuracy of vertical root fracture diagnosis [17]. The result of this study is different from all of the previously mentioned studies, which can be due to the use of different algorithms. The CBCT algorithm and device used in Oliveira’s study were from KaVo, which is different from the present study. In addition, the diagnosis of vertical root fracture in CBCT images is already a challenging diagnostic task.

In 2022, Gomes et al. investigated the effect of the artifact reduction algorithm in teeth restored with different posts and found that in the case of nickel–chromium and chrome–cobalt posts, these algorithms did not significantly reduce the amount of artifacts. However, in the case of silver–palladium posts, the effect of the MAR algorithm was significant [23]. This indicates that the material and alloy of the restoration were a significant factor in the effectiveness of MAR algorithms.

In the field of orthodontics, the effectiveness of MAR algorithms has not extensively studied. In a study by Isman et al. in 2019, 40 premolar and molar teeth without cavities were selected. Three types of stainless steel, ceramic, and plastic brackets and two types of stainless steel and titanium arch wires were attached to the teeth, and CBCT images were taken with a small and large field of view, with or without MAR activation. The authors concluded that titanium and stainless steel arch wires and stainless steel brackets alone or with stainless steel wires make it impossible to detect caries without interproximal cavities, but the field of view and artifact reduction mode have no effect on the detection of these caries [18]. This qualitative study had results confirming the quantitative findings of the present study. It also points out the detrimental effects of artifacts arising from metallic objects on proper diagnosis.

In 2023, McLaughlin et al. used pig cadaver mandibles to investigate the effects of MAR algorithms on the accuracy of temporary anchorage devices–tooth root contact. Their findings indicate that the application of a currently available Planmeca MAR algorithm did not lead to a significant reduction in the rate of a false positive diagnosis of contact. They suggested that further optimization of the MAR algorithm for this purpose may be needed [19].

In this study, the metal artifacts were quantitatively analyzed in the palatal region, as impacted canines are most often in a palatal position compared to the erupted dentition [28]. Additionally, the central regions of the CBCT image are less severely affected by down-sampling artifacts, including aliasing, compared with the peripheral regions [29]. The CNR equation uses the mean and standard deviation of voxel values in a homogeneous material, such as water in the region of interest and a control region unaffected by introduced artifacts. Even in the control region, which is by definition unaffected by the artifacts in question, all voxels do not possess the same value due to inherent image noise. Therefore, the CNR equation considers inherent variations in voxel values as well as the introduced voxel value variability.

The findings of this study are limited by its in vitro design. Similar studies on CBCT images of patients with orthodontic brackets and wires are suggested, although standardization of the conditions of the scans and metallic objects would be challenging. Additionally, the effect of bracket and wire material on the effectiveness of MAR algorithms can be investigated. The studies on the effectiveness of MAR algorithms are also limited by the lack of complete knowledge of the mechanism of action of such algorithms, as they are not revealed by the manufacturers. Further research studies using brackets and wires of different materials are recommended. Understanding the effects of MAR algorithms, as well as the presence of metallic objects in the CBCT field of view on images, may help operators to reconsider pre-scan patient preparation procedures including removal of orthodontic wires and de-bonding orthodontic brackets.

## 5. Conclusions

According to the results of this study, MAR activation had a significant impact in increasing CNR, but only when the brackets were in a palatal position. In the other bracket and wire positions, this increase was not significant. However, the slight increase in quantitative measures of image quality might be clinically important and justify the application of MAR algorithms when scanning arches with orthodontic appliances.

## Figures and Tables

**Figure 1 diagnostics-14-00159-f001:**
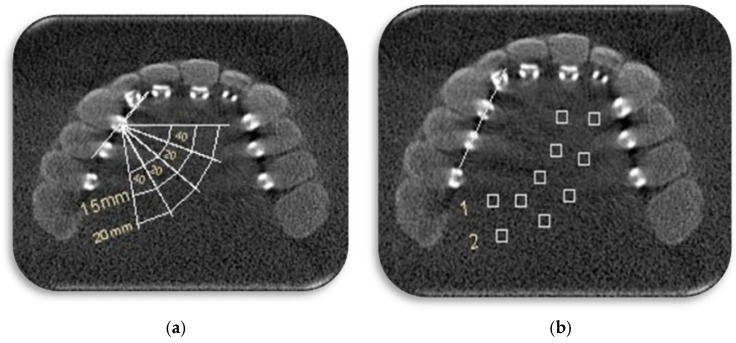
(**a**) A line was drawn from the center of the bracket along the buccal surface of the tooth. A line perpendicular to this line was drawn. Regions of interests (ROIs) located at intervals of 15 and 20 mm from center of bracket and with angles of 20 and 40 degrees on each side of perpendicular line and also on the perpendicular line. (**b**) 10 ROIs (boxes) of the same size (2 mm × 2 mm) were established for brackets: 5 ROIs in the 15 mm distance (1) and 5 ROIs in the 20 mm distance (2).

**Table 1 diagnostics-14-00159-t001:** Mean and standard deviation (SD) of contrast-to-noise ratio (CNR) in the two metal artifact reduction (MAR) modes.

MAR Mode	Mean (SD)	*p*-Value
On	0.22 (0.107)	0.220
Off	0.20 (0.118)	
Total	0.21 (0.113)	

**Table 2 diagnostics-14-00159-t002:** Mean and standard deviation (SD) of contrast-to-noise ratio (CNR) in 15 and 20 mm distances from the center of the brackets in the two metal artifact reduction (MAR) modes.

Distance	MAR Mode	Mean (SD)	*p*-Value
15 mm	On	0.24 (0.127)	0.657
	Off	0.23 (0.142)	
	Total	0.24 (0.134)	
20 mm	On	0.20 (0.080)	0.092
	Off	0.16 (0.079)	
	Total	0.18 (0.081)	

**Table 3 diagnostics-14-00159-t003:** Mean and standard deviation (SD) of contrast-to-noise ratio (CNR) in different teeth in the two metal artifact reduction (MAR) modes.

Tooth	MAR Mode	Mean (SD)	*p*-Value
Central incisor	On	0.25 (0.140)	0.674
	Off	0.22 (0.152)	
	Total	0.23 (0.143)	
Lateral incisor	On	0.25 (0.105)	0.765
	Off	0.23 (0.123)	
	Total	0.24 (0.111)	
Canine	On	0.21 (0.088)	0.823
	Off	0.20 (0.117)	
	Total	0.20 (0.101)	
First premolar	On	0.20 (0.118)	0.559
	Off	0.17 (0.130)	
	Total	0.18 (0.122)	
Second premolar	On	0.22 (0.090)	0.138
	Off	0.17 (0.056)	
	Total	0.20 (0.078)	

**Table 4 diagnostics-14-00159-t004:** Mean and standard deviation (SD) of contrast-to-noise ratio (CNR) in different bracket and wire positions in the two metal artifact reduction (MAR) modes.

Bracket and Wire Position	MAR Mode	Mean (SD)	*p*-Value
B-B	on	0.18 (0.077)	0.230
	off	0.14 (0.063)	
	Total	0.16 (0.071)	
BW-B	on	0.34 (0.125)	0.969
	off	0.33 (0.158)	
	Total	0.34 (0.138)	
B-P	on	0.22 (0.107)	0.030
	off	0.13 (0.063)	
	Total	0.17 (0.098)	
BW-P	on	0.22 (0.044)	1.000
	off	0.22 (0.042)	
	Total	0.22 (0.042)	
No-BW	on	0.18 (0.090)	0.907
	off	0.17 (0.098)	
	Total	0.17 (0.092)	

## Data Availability

Data are contained within the article.

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
