# Peer review of "A Quantitative Evaluation of the Effectiveness of the Metal Artifact Reduction Algorithm in Cone Beam Computed Tomographic Images with Stainless Steel Orthodontic Brackets and Arch Wires: An Ex Vivo Study"

_diagnostics, 2024, doi:10.3390/diagnostics14020159_

Round 1

Reviewer 1 Report

Comments and Suggestions for Authors

In my opinion, the research topic may be truly interesting not only for orthodontists and dentists but also for radiologists and every specialist involved in the diagnostic of maxillo-facial structures. 

However, there are several lacks and weaknesses in the language, paper structure and  methodology that should be considered, for example: 

- following the PICO guidelines, some information is missing in the title, i.e. the study design, the sample groups and comparators;

- it is not clear the rationale of the study, please add more dat on previous published papers; 

- please, specify the different aims in primary and secondary ones reporting the same in the results / conclusions;

- about the statement "for evaluation orthodontic patients. However, these radiographs 51 are limited by their two-dimensional nature and thus cannot effectively visualize the re-52 lationships of different structures." 'please, add some indications in doing CBCT for orthodontic reasons; 

- the lines from 60 to 70 did not clearly and easily describe the artifacts, which seem to be 4 plus 2. Please, reword; 

- in lines from 90 to 94 redundant sentences are reported; 

- in the discussion there are similar concepts reported in the Introduction; 

- the references list should be improved by adding previous papers to justify the research topic and design, as mentioned above. 

Comments on the Quality of English Language

The English language should be improved by a native speaker because it presents several redundancies, typos, and incorrect sentences.

Author Response

We would like to express our gratitude to you for your helpful and constructive comments. We appreciate the points noted by the reviewers and think that by addressing them, our manuscript has improved significantly. We have highlighted the changes made in the manuscript. In the following section, we provide a point-by-point reply to the reviewers’ comments:

  • following the PICO guidelines, some information is missing in the title, i.e. the study design, the sample groups and comparators;

Including all these items would make for a long title. We have added the study design (page 1, lines 4,5)

  • it is not clear the rationale of the study, please add more dat on previous published papers; 

We believe that a lack of studies investigating the performance of MAR algorithms for the condition in question (brackets and wires in buccal and lingual position) is a viable rationale. We have explored further in the introduction (pages 2,3, lines 95-100, 104-106).

  • please, specify the different aims in primary and secondary ones reporting the same in the results / conclusions;

We have revised the aims based on the reviewers comments (page 3, lines 106-110).

  • about the statement "for evaluation orthodontic patients. However, these radiographs 51 are limited by their two-dimensional nature and thus cannot effectively visualize the re-52 lationships of different structures." 'please, add some indications in doing CBCT for orthodontic reasons; 

We have now added some applications of CBCT imaging in the field of orthodontics (page 2, lines 61-64).

  • the lines from 60 to 70 did not clearly and easily describe the artifacts, which seem to be 4 plus 2. Please, reword; 

As mentioned, artifacts in CBCT images can be classified into 4 groups. We have now added more explanations (page 2, lines 68-71).

  • in lines from 90 to 94 redundant sentences are reported; 

Edited.

  • in the discussion there are similar concepts reported in the Introduction; 

We have removed the redundant concepts.

  • the references list should be improved by adding previous papers to justify the research topic and design, as mentioned above. 

We have improved the reference list based on this comment.

Reviewer 2 Report

Comments and Suggestions for Authors

The manuscript describes the study of the quantitative evaluation of the effectiveness of MAR algorithm in CBCT images of teeth with stainless steel (SS) orthodontic brackets with and without arch-wire in buccal and lingual positions.

The topic is not novel, there is large body of similar literature; the scope and methods are limited.

MAR developed by different producers are different, thus it is paramount to specify upfront which CBCT machine was used and which MAR.

The problem statement is good “no quantitative study has been performed on the performance of CBCT MAR algorithms in reduction of artifacts caused by brackets and archwires located in buccal and lingual positions” and the authors stated “Orthodontic brackets can be made of metal, such as stainless steel and titanium brackets, or non-metallic materials, such as plastic, ceramic, and plastic composite brackets…” but studying only SS brackets and archwire significantly limits the scope of the study. Why did you not study titanium?

Since “Objects with high density and atomic number can cause metal artifacts in CBCT images,” are we more concerned with materials of different densities or objects with different geometry?

Another function of MAR is to reduce the voxel value variability. However, the authors only reported CNR, and the equation of CNR appears to cancel out the part of voxel value variability. Can you also report if MAR reduced the voxel value variability or not.

Extensive English editing is necessary.

The Conclusion is overly brief. Expand it and add significance statement.

Should be “orthodontic brackets with and without arch-wire”

Comments on the Quality of English Language

extensive editing

Author Response

We would like to express our gratitude to you for your helpful and constructive comments. We appreciate the points noted by the reviewers and think that by addressing them, our manuscript has improved significantly. We have highlighted the changes made in the manuscript. In the following section, we provide a point-by-point reply to the reviewers’ comments:

  • MAR developed by different producers are different, thus it is paramount to specify upfront which CBCT machine was used and which MAR.

We agree with this comment. We have now added the relevant information to the abstract and introduction as well (page 1, line 22, page 3, lines 108, 109).

  • The problem statement is good “no quantitative study has been performed on the performance of CBCT MAR algorithms in reduction of artifacts caused by brackets and archwires located in buccal and lingual positions” and the authors stated “Orthodontic brackets can be made of metal, such as stainless steel and titanium brackets, or non-metallic materials, such as plastic, ceramic, and plastic composite brackets…” but studying only SS brackets and archwire significantly limits the scope of the study. Why did you not study titanium?

As evident in the revised objectives, the main question was the amount of artifacts when brackets with/without wires are placed in either buccal or lingual positions. Therefore, adding different materials of wire and brackets would have made too much variable and would require further research as mentioned in the suggestions for future studies.

  • Since “Objects with high density and atomic number can cause metal artifacts in CBCT images,” are we more concerned with materials of different densities or objects with different geometry?

High-density materials are generally those with a high atomic number.

  • Another function of MAR is to reduce the voxel value variability. However, the authors only reported CNR, and the equation of CNR appears to cancel out the part of voxel value variability. Can you also report if MAR reduced the voxel value variability or not.

CNR is calculated in a homogenous material, such as water. In the formula used for the calculation of CNR, it accounts for voxel value variability, i.e. noise. However, for further clarification, we have added a section regarding this topic to the discussion (pages 8, lines 305-310)

  • Extensive English editing is necessary.

We have edited the text for grammatical errors and typos.

  • The Conclusion is overly brief. Expand it and add significance statement.

We have added clinical implications to the conclusion (page 8, lines 325-327)

  • Should be “orthodontic brackets with and without arch-wire”

We could not find this statement. However, we have revised the manuscript text. If the issue persists, please let us know with page and line for better localization.

Reviewer 3 Report

Comments and Suggestions for Authors

This study presents a potential well-designed investigation aimed at evaluating the efficacy of a metal artifact reduction algorithm in CBCT images in the presence of stainless steel orthodontic brackets and arch-wires.

I have some comments on it:

1. The introduction section should be updated with more relevant and newer articles..

2. The discussion part is not clear and must be rewritten with more pertinent article and fluent language.

  Comments on the Quality of English Language

Moderate editing of English language required.

Author Response

We would like to express our gratitude to you for your helpful and constructive comments. We appreciate the points noted by the reviewers and think that by addressing them, our manuscript has improved significantly. We have highlighted the changes made in the manuscript. In the following section, we provide a point-by-point reply to the reviewers’ comments:\

  • The introduction section should be updated with more relevant and newer articles.

We have improved the introduction with newer articles relevant to orthodontics (pages 2,3).

  • The discussion part is not clear and must be rewritten with more pertinent article and fluent language.

We have improved the discussion (pages 6-8).

Round 2

Reviewer 1 Report

Comments and Suggestions for Authors

Thank you for your review. 

Good luck!

Comments on the Quality of English Language

The English language could always be improved by a native speaker.  

Reviewer 2 Report

Comments and Suggestions for Authors

change the "with or without" to and throughout since both were studied.